# The Value of Adding Surveillance Cultures to Fluoroquinolone Prophylaxis in the Management of Multiresistant Gram Negative Bacterial Infections in Acute Myeloid Leukemia

**DOI:** 10.3390/jcm8111985

**Published:** 2019-11-15

**Authors:** Christelle Castañón, Ahinoa Fernández Moreno, Ana María Fernández Verdugo, Javier Fernández, Carmen Martínez Ortega, Miguel Alaguero, Concepción Nicolás, Laura Vilorio Marqués, Teresa Bernal

**Affiliations:** 1Servicio de Hematología, Hospital Universitario Central de Asturias, 33011 Oviedo, Spain; chriscasta@gmail.com (C.C.); aftorozo@gmail.com (A.F.M.); conchanicolas61@gmail.com (C.N.); 2Instituto de Investigación Sanitaria del Principado de Asturias, 33012 Oviedo, Spain; ANAMARIA.FERNANDEZVERDUGO@asturias.org (A.M.F.V.); javifdom@gmail.com (J.F.); lvilm@unileon.es (L.V.M.); 3Servicio de Microbiología, Hospital Universitario Central de Asturias, 33011 Oviedo, Spain; 4Servicio de Medicina Preventiva y Salud Pública, Hospital Valle del Nalón, 33920 Langreo, Spain; mamenpreventiva@gmail.com; 5Área de Gestión Clínica de Farmacia, Hospital Universitario Central de Asturias, 33011 Oviedo, Spain; alaguerom@gmail.com; 6Departamento de Medicina, Universidad de Oviedo, 33011 Oviedo, Spain; 7CIBER-Enfermedades Respiratorias. Instituto de Salud Carlos III, 28029 Madrid, Spain

**Keywords:** acute myeloid leukemia, multi-resistant Gram-negative bacteria, quinolone prophylaxis, surveillance cultures, stem cell transplantation

## Abstract

Multidrug resistant Gram-Negative Bacterial Infections (MR-GNBI) are an increasing cause of mortality in acute myeloid leukemia (AML), compromising the success of antineoplastic therapy. We prospectively explored a novel strategy, including mandatory fluoroquinolone prophylaxis, weekly surveillance cultures (SC) and targeted antimicrobial therapy for febrile neutropenia, aimed to reduce infectious mortality due to MR-GNBI. Over 146 cycles of chemotherapy, cumulative incidence of colonization was 50%. Half of the colonizations occurred in the consolidation phase of treatment. Application of this strategy led to a significant reduction in the incidence of GNB and carbapenemase-producing *Klebisella pneumoniae* (cpKp) species, resulting in a reduction of infectious mortality (HR 0.35 [95%, CI 0.13–0.96], *p* = 0.042). In multivariate analysis, fluroquinolone prophylaxis in addition to SC was associated with improved survival (OR 0.55 [95% CI 0.38–0.79], *p* = 0.001). Targeted therapy for colonized patients did not overcome the risk of death once cpKp or XDR *Pseudomonas aeruginosa* infections were developed. Mortality rate after transplant was similar between colonized and not colonized patients. However only 9% of transplanted patients were colonized by cpkp. In conclusion, colonization is a common phenomenon, not limited to the induction phase. This strategy reduces infectious mortality by lowering the global incidence of GN infections and the spread of resistant species.

## 1. Introduction

Half of the deaths due to Acute Myeloid Leukemia (AML) are related to infections, being Gram negative bacteria the most prominent pathogens [1]. The increasing appearance of multidrug resistant Gram negative bacterial infections (MR-GNBI) constitutes one the most important medical threats of present time [2] and has a profound impact in the treatment of AML. On one hand, a high rate of complications and death associated to them have been reported [3,4]. On the other, MR-GNBI negatively impact the delivery and success of chemotherapy and stem cell transplantation (SCT) [5].

The best strategy to face this problematic situation is yet to be defined. The use of fluoroquinolone prophylaxis (FP) in endemic settings is a matter of debate, as they result in an increasing prevalence of resistant microorganisms and may reduce the efficacy of subsequent antibacterial treatment [6]. Routine active surveillance cultures (SC) from gastrointestinal tract and contact precautions in colonized patients have proven to be useful in intensive care and solid organ transplantation settings [7,8] but its cost-effectiveness in AML is to be demonstrated [9]. Furthermore, there is no international agreement on how to organize and implement active surveillance control measures for the detection of colonized AML [10]. 

In order to clarify the actual role of FP and SC in the management of infection in AML patients, we prospectively evaluated the efficacy of an active surveillance program (ASP) on the reduction of infection and mortality due to multiresistant Gram negative bacteria in an endemic MR-GNB setting.

## 2. Methods 

We designed a prospective study conducted from July 2014 to January 2019. The Institutional Review Board approved the study, and informed consent was obtained from patients. All AML consecutive patients, older than 18 years, who were treated with intensive chemotherapy at Hospital Universitario Central de Asturias were included in the study. Patients treated with hypomethylating agents were excluded, as the infectious complications associated with these drugs are different to those that follow conventional chemotherapy [11]. The ASP was started in May 2016. Therefore, the whole cohort was split in two subgroups: cohort 1 comprises the period before implantation, whereas cohort 2, the period after the onset of the measures. The primary end point of the study was infectious mortality. Secondary end points included incidence of colonization and infection by MR-GNB and overall survival.

### 2.1. Intervention

Cohort 1 included all AML patients who were treated with intensive chemotherapy and in whom SC where not routinely performed. FP, although suggested, was decided at the discretion of the physician in charge. This practice was allowed as the percentage of fluoroquinolone resistance among Gram negative bacteria in our setting ranged from 50 to 70% depending on the species, and efficacy of FP under these situations is not clear [12]. 

ASP (cohort 2) included the following actions: (1) Systematic FP from the beginning of chemotherapy; (2) rectal swabs for MR-GNB surveillance collection from the first day of hospital admission and weekly thereafter; (3) Contact precautions for MR-GNB colonized and infected patients; (4) targeted therapy for febrile neutropenia in colonized patients according microbiologic results of surveillance cultures. Antibacterial therapy was considered adequate if it included at least one drug displaying activity against the isolated microorganism.

For both cohorts (all patients in cohort 2 and when indicated in cohort 1), FP consisted on ciprofloxacin 500 mg PO q12 hours starting on day one from the chemotherapy cycle and continuing until absolute neutrophil count > 0.5 × 10^9^/L. Contact precautions for MR-GNB infected patients included patient placement (single-room), gowns, gloves, noncritical patient-care equipment/patient-dedicated use of such equipment, enhanced environmental measures (water filters and cleaning) and antiseptic baths.

### 2.2. Data Collection

Recorded data included age, gender, phase of the disease, hospitalization period, white blood cell count at diagnosis, cytogenetic and molecular testing [13], comorbidities according the Hematopoietic Comorbidity Index [14], date and type of chemotherapy, duration of neutropenia, date of allogeneic stem cell transplantation, relapse and antimicrobial exposure.

According to protocol, blood, urine cultures and cultures from sites of infection were acquired at the onset of all febrile episodes. Infection was defined as isolation of Gram negative bacteria from blood and other sterile or non-sterile body sites associated to compatible signs [15]. Onset of MR-GNB infection was defined as the day when the first positive culture was taken. 

Microbiological samples were processed in the Microbiology laboratory, according to the type of sample received. Bacterial identification was performed by MALDI-TOF MS (Microflex™; Bruker Daltonik GmbH, Bremen Germany) and antimicrobial susceptibility testing of suspicious colonies was carried out by the Microscan system (Beckman Coulter, CA, USA) and interpreted according to the last breakpoints proposed by the European Committee on Antimicrobial Susceptibility Testing [16].

The microorganisms that were routinely analyzed in surveillance samples were carbapenem-resistant *Acinetobacter baumannii*, extended spectrum β-lactamase (ESBL), carbapenemase-producing *Enterobacteriaceae* and extremely drug resistant (XDR) *Pseudomonas aeruginosa.*

### 2.3. Statistical Analysis

Data are presented as median (interquartile range). Univariate comparisons were done using the chi square test (categorical variables), Fisher’s exact test (for contingency tables with frequencies below 5) or Wilcoxon test (for continuous variables). 

Cumulative incidence of colonization and infection by a MR-GNB were calculated accounting for the competing risk of death and censoring at the time of transplantation. A similar competitive model was developed to quantify infection free survival (IFS) and death rates secondary to infectious and non-infectious causes. IFS was defined as the length of time that patients survived without an infection. In patients with partial response, stable or progressive disease, death was attributed to infection if it was the result of an acute event involving sites of infection in absence of other causes contributing to death. Additionally, mortality rate 4 months after SCT was assessed. Factors involved in overall survival were analyzed using a multivariate Cox proportional hazard model. All the statistical analyses were performed using the R statistical package (version 3.2.1).

## 3. Results

Overall, 102 patients received 233 chemotherapy cycles. Thirty-seven patients were included in cohort 1 and 65 in cohort 2. As per design, all patients received an induction cycle of intensive chemotherapy. A second induction cycle was needed in seven and 11 patients in cohorts 1 and 2, respectively. Consolidation chemotherapy was administered to those reaching complete remission (26 patients in cohort 1 and 38 patients in cohort 2 received one consolidation cycle, 13 and 20 patients received two consolidation cycles, and four and 12 patients received three cycles consolidation respectively). Six patients received a re-induction cycle for relapse occurring 24, 15, 12, 10, 9 and 7 months after the first complete remission. Gut colonization by MR-GNB was ruled out in five of them before initiating chemotherapy.

Seventy-seven out of 102 patients achieved complete remission, being the overall CR rate 76%, with no differences between cohorts: 77 and 73% in cohort 1 and 2, respectively (*p* = 0.7). Allogeneic stem cell transplant was performed in 31% of patients. No significant differences regarding demographic and leukemia-associated variables were observed between both cohorts (Table 1).

### 3.1. Compliance with the Active Surveillance Program 

Hospitalization was mandatory from the first day of chemotherapy until the recovery of neutropenia, regardless colonization or remission status and cycle of chemotherapy (induction or consolidation). Likewise, when patients were discharged from hospital at hematologic recovery, individual isolation waiting and examining rooms were also provided. These actions ensured that contact precautions were observed in all colonized patients during the whole duration of study period. Length of hospitalization was similar in the two periods, with 47 days (30–70) of hospital stay in cohort 1 compared to 60 days (40–66) in cohort 2 (*p* = 0.2).

Proportion of patients receiving FP increased from an average value of 29% in cohort 1 up to 87% in cohort 2, with the exception of induction 1, in which more patients were treated with broader spectrum antibiotics because of fever at presentation of leukemia (Table 1). 

Weekly SC for MR-GNB were performed in 88% (57/65) of patients. In the remaining eight patients, the time interval between consecutive SC was longer than a week.

Antimicrobial therapy in patients colonized by a MR-GNB at the onset febrile episodes was considered adequate in 87% (35/40) of the episodes, and consisted of carbapenems in 24/35 (68%) or colistin in 11/35 (32%) of the prescriptions. Piperacillin-tazobactam, and cephalosporines were considered inadequate empiric therapy (in two and three patients, respectively). 

### 3.2. Colonization

Cumulative incidence of colonization is shown in Figure 1. Rate of colonization by a MR-GNB at 5 months from the beginning of treatment was 50% (95 confidence interval 39–65%). The proportion of new colonizations in subsequent chemotherapy cycles was 18/65 (28%), 1/5 (20%), 9/38 (24%), 2/20 (10%) and 2/12 (16%) for induction 1, induction 2 and consolidation 1, 2 and 3 respectively.

The most frequent microorganism responsible for colonization was ESBL-producing *K. pneumoniae* followed by XDR *P. aeruginosa* and ESBL- and carbapenemase-producing *Enterobacter cloacae*. Two different microorganisms colonized 3 patients simultaneously during induction 1 (two patients with ESBL-producing *K. pneumoniae* and XDR *P. aeruginosa*, and a single patient with ESBL-producing *K. pneumoniae* and carbapenem-resistant *A. baumannii*). Bacterial isolates recovered and their susceptibility patterns are detailed in Table 2. 

### 3.3. Infection

Two hundred and thirty-six febrile episodes affected the whole population. Types of infections are detailed in Table 3. One hundred and forty-two cases (60%) were microbiologically documented. To compare the incidence of microbiologically documented infections between the induction and consolidation phases we calculated the rate of infections occurring during induction or consolidation cycles divided by the total number of chemotherapy cycles administered in each phase. Rate of infection was 1.05 and 2 during induction and consolidation in cohort 1 compared to 0.89 and 1.14 in cohort 2. 

Gram negative bacteria were isolated in 56 (39%) of the febrile episodes. A lower incidence of Gram-negative infections was observed during cohort 2, as shown in Figure 2A. This difference was mainly attributable to a lower proportion of Gram-negative isolates during the consolidation cycles of chemotherapy. There were 20/27 (74%) Gram negative infections in cohort 1 compared to 18/44 (41%) in cohort 2 (*p* = 0.006). On the contrary, rate of Gram positive infections significantly increased during consolidation in cohort 2 compared to cohort 1.

When taking into account the competing risk of death, no differences in cumulative incidence of infection by a MR-GNB were observed (*p* = 0.78 in the competing risk model, Figure 2B). At 5 months from the beginning of antileukemic treatment, infection rates were 30% (95% CI 18–49) in cohort 1 and 32% (95% CI 22–46%) in cohort 2. MR-GNB responsible for infection and their susceptibility pattern are shown in Table 4. Changes in antibiotic susceptibility of isolated microorganisms were observed between periods. In cohort 2, a trend to a reduction in carbapenemase-producing *K*. *pneumoniae* (cpKp) strains was observed, *p* = 0.06. 

Sixty percent of colonized patients developed infection. The same species that previously colonized patients were identified in 74% of the infectious episodes. 

### 3.4. Survival

With a median follow up of 150 days (98–289), median overall survival was 25,2 months (8,2-NA) without differences between cohorts (Log Rank *p* = 0.4, Figure 3A). Infection free survival, estimated using a competing risks model, was also similar in both cohorts, (HR 0.92 [0.54–1.58], *p* = 0.77, Figure 3B).

When causes of death were analyzed, cumulative incidence of death secondary to infection was lower in cohort 2 (HR 0.35 [0.13–0.96], *p* = 0.042, Figure 3C), with no difference in the cumulative incidence of non-infectious deaths (HR 1.45 [0.55–3.82], *p* = 0.45, Figure 3D).

Survival was not different between infected patients by a MR-GNBI and non-infected patients (HR 0.99 [0.50–1.97], *p* = 0.97, Figure 4A). Among patients infected with MR-GNB, those infected by cpKp or XDR *P. aeruginosa* showed a significantly increased mortality (HR 9.5 [1.2–75], *p* = 0.032, Figure 4B). These results were similar in both cohorts.

Regarding transplanted patients, mortality rate 4 months after transplant was not different between colonized and non-colonized patients (88% vs 91%, *p* = 0.5). Of note, none of the transplanted patients was colonized by *P. aeruginosa* XDR and only 1 (9%) was colonized by cpKp. Conversely, among younger, non-transplanted patients, cpKp and XDR *P. aeruginosa* represented 40% of the colonizing microorganisms.

### 3.5. Targeted Therapy

Infection was the cause of death in 10/17 patients in cohort-1. XDR *P. aeruginosa* and *cpKp* were isolated in three patients each. Simultaneous infection by XDR *P. aeruginosa* and cpKp was detected in one of them. Five of the patients with MR-GNBI were treated with piperacillin-tazobactam at the beginning of febrile neutropenia. Five of these deaths occurred in the aplastic phase following consolidation cycles. In cohort 2, infection accounted for 6/20 (30%) of deaths, being cpKp and XDR *P. aeruginosa* the cause of 2. These patients were previously colonized by the same microorganisms and received targeted treatment according the susceptibility pattern of the isolated microorganism. 

In order to assess the specific impact of each of factor on survival, a multivariate Cox proportional hazards model was designed including age, comorbidity index, genetic risk, FP and period of study, with an interaction between these last 2 factors. FP significantly decreased mortality only in cohort 2 (Table 5).

## 4. Discussion

In this prospective study we have analyzed the impact of an active surveillance program on infectious mortality in a cohort of AML patients treated in a high-endemic MR-GNB setting [17,18,19]. Our results show that strict compliance with fluoroquinolone prophylaxis, together with systematic surveillance cultures and contact precautions for colonized patients by MR-GNB, resulted in a reduction in the incidence of MR-GNBI and infectious mortality during the aplastic period of chemotherapy. A parallel increase in Gram positive infections was observed. Additionally, we report information about the natural history of colonization by MDR-GNB through the whole treatment of AML until allogeneic stem cell transplant.

Traditionally, the risk of infection in AML has been considered higher during induction, compared to the consolidation phase, due to the uncontrolled status of the disease. However, in the last years, an increasing incidence in MR-GNB infections during the consolidation cycles of chemotherapy has been reported, associated to high mortality rate [20,21]. This finding highlights the need to redefine the infectious risk of AML in the time of multidrug resistance and to search new ways of infection control also during these phases. 

In our study, the systematic use of fluoroquinolone prophylaxis has resulted in a reduction in the global incidence of GNB infections during the consolidation cycles. These results apparently contradict previous observations reporting an increasing incidence in MR-GNBI during consolidation attributable to prophylaxis [20,21]. A deeper analysis revealed a redistribution of resistant strains within the same species, in such a way that the incidence of carbapenemase-producing strains was reduced in parallel to an increase in ESBL-producing *K. pneumoniae*. Contact precautions of colonized patients may have contributed to limiting the spread of cpKp strains, as cross transmission is the main mechanism involved in the spread of these species [10,22]. The final consequence of this epidemiological trend was a reduction in infectious mortality due to a lower incidence of carbapenemase-producing *Enterobacteriaceae* infections. This finding is not surprising, as carbapenemase-producing strains are known for their high mortality in neutropenic patients [3,4,23] and is in line with previous reports showing no increase in the risk of mortality due to infections by ESBL-producing *Enterobacteriaceae* [23]. Our report agrees with previous ones showing that fluoroquinolone-induced bacterial resistance does not impact negatively on mortality [24], and supports the recommendations of current guidelines [12,15]. In line with this finding, we did not observe an increase mortality after the early postransplant period among colonized patients, probably due to the limited virulence of these strains.

An interesting finding was the increase in the rate of microbiologically documented infections and Gram positive infections during consolidation compared to induction. A similar trend has been reported previously associated to the use of indwelling catheters and high dose Ara-C, especially in patients without fluoroquinolone prophylaxis [21,25]. The findings of our study support this hypothesis, since in our center both the use of indwelling catheters and high dose Ara-C during consolidation are standard approaches for the treatment of AML patients. Moreover, the highest rate of infection/cycle during consolidation was observed in cohort 1, where most of the patients did not received fluoroquinolone prophylaxis.

AML patients showed a high rate of colonization by MR-GNB, being *K. pneumoniae* the most frequent species. Importantly, the results show that colonization is an ongoing phenomenon, occurring through the whole treatment of the disease. In consequence, limiting the surveillance cultures to the induction phase would lead to lose crucial information regarding actual colonization status, and to erroneously dismiss contact precautions. In fact, another potential explanation for the observed reduction in MR-GNB incidence may be that, having found new colonized patients beyond the first induction cycle, contact precautions could be implemented earlier.

As colonization is a risk factor for subsequent infection by the same species [23], an additional benefit of performing surveillance cultures is to target the antimicrobial therapy at the onset of febrile neutropenia. Potential benefits of this strategy include the reduction in the overuse of broad spectrum antibiotics for the treatment of febrile neutropenia in non-colonized patients and the delivery of effective antibiotics for those colonized, avoiding the mortality associated with inadequacy of empirical therapy [26,27]. In our study, adequate antibiotics were prescribed for more than 80% of the colonized patients. In contrast, in the previous period, seventy percent of patients who dead as a consequence of MR-GNBI had received inadequate therapy. In spite of adequacy, mortality remained high once cpKp *or* XDR *Pseudomonas aeruginosa* infections were developed. A delay in the beginning of antibiotics was discarded as the cause of this high mortality, since all patients in cohort 2 remained hospitalized from the first day of chemotherapy until the recovery of neutropenia regardless of the phase of the disease. Optimization of antimicrobial therapy according to colonization status using combination of existing antibiotics or new ones, should be explored to improve the outcome of febrile neutropenia in those patients colonized by highly resistant species. 

The main limitation of our study is the low number of patients and its unicentric nature, which lowered the statistical power of the results and precludes their generalization. However, the epidemiology of our center, rate of colonization during induction and the trends in mortality are similar to previous reports [4,26]. In spite of these limitations, this is the first study to show a detailed analysis of the process of colonization in a highly homogeneous cohort of AML patients through the whole period of treatment and to demonstrate the feasibility of performing surveillance cultures for the MR-GNB infection management. Additionally, the results support the addition of fluoroquinolone prophylaxis in addition to contact precautions and surveillance cultures as a way to reduce the incidence and spread of Gram-negative infections, including carbapenemase-producing *Enterobacteriaceae,* and targeting antibacterial therapy in colonized patients for the treatment of febrile neutropenia. 

## 5. Conclusions

Our results show that for FP to be effective in AML patients, it must be included in a set of measures including systematic surveillance cultures and isolation protocols. Microbiological data obtained from surveillance cultures allow the early onset of effective contact precautions and targeting therapy in patients with multiresistant Gram negative bacteria. This strategy must be continued throughout every chemotherapy cycle and during stem cell transplantation. Novel strategies to prevent and control infectious episodes by highly resistant microorganisms need to be explored.

## Figures and Tables

**Figure 1 jcm-08-01985-f001:**
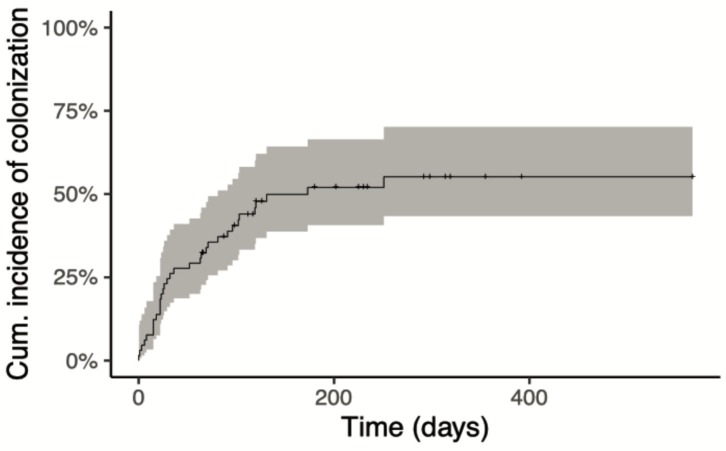
Cumulative incidence of colonization by multi-resistant Gram-negative bacteria.

**Figure 2 jcm-08-01985-f002:**
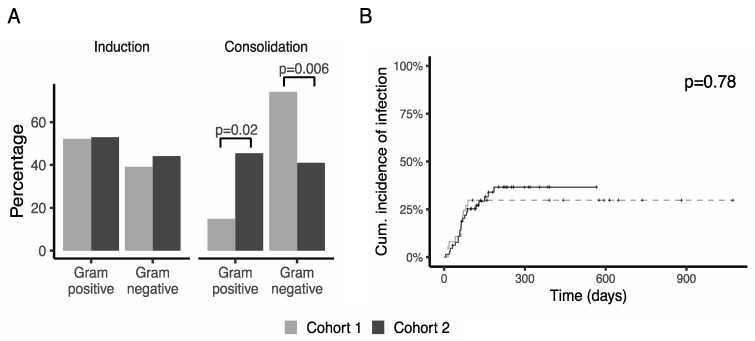
Incidence of infection. (**A**) Distribution of Gram-positive and Gram-negative infections during induction and consolidation cycles in both cohorts. A significant reduction in Gram-negative bacterial infections during consolidation cycles was observed in cohort 2. (**B**) Cumulative incidence of infection by multi-resistant Gram-negative bacteria, accounting for the competing risk of death.

**Figure 3 jcm-08-01985-f003:**
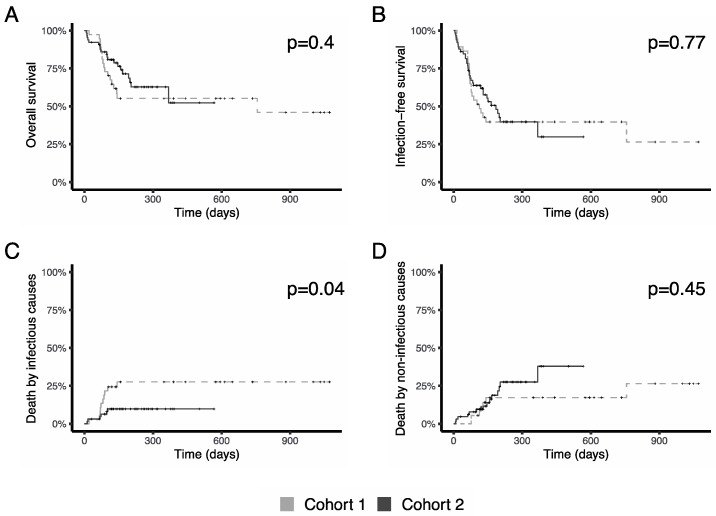
Survival. (**A**) Comparison of overall survival between cohorts. (**B**) Infection-free survival: there are no differences in the time that patients survived without an infection between cohorts. (**C**) Death rates. A significant decrease in deaths secondary to infections were observed in cohort 2. (**D**) Cumulative incidence of non-infectious deaths.

**Figure 4 jcm-08-01985-f004:**
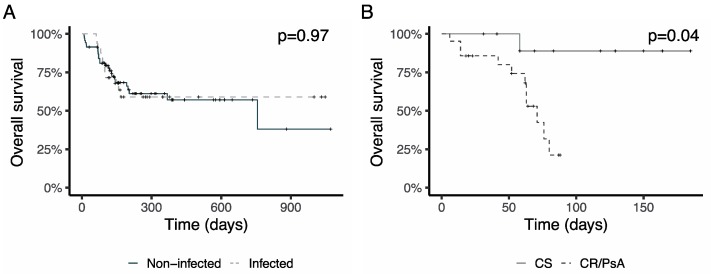
Impact of multiresistant Gram negative infection (MR-GNBI) on survival. (**A**) Overall survival of patients infected by a MR-GNBI compared to non-infected patients. (**B**) Mortality rate in patients infected by carbapenemase-producing *K. pneumoniae* (CR) or extremely resistant *P. aeruginosa* (PsA) compared to other carbapenem-sensitive (CS) Gram-negative infections.

**Table 1 jcm-08-01985-t001:** Patients´ characteristics and univariate comparisons between cohorts. Data are expressed as number and percentage and as median and interquartile range.

	Cohort 1	Cohort 2	*p*
Number of patients	37	65	
Number of cycles	87	146	
Age, mean years (range)	58 (44–64)	58 (50–64)	0.5
Male sex N (%)	16 (43)	30 (46)	0.9
Prognostic group N (%)			
Good	15 (40)	28 (43)	0.8
Intermediate	11 (30)	16 (25)	
Poor	11 (30)	21 (32)	
White Blood Cell Count × 10^9^/L (range)	10.07 (3.64–31.6)	8.07 (2.43–21.9)	0.2
Comorbidity index (%)			
0–2	32 (86)	53 (82)	0.6
>2	5 (14)	12 (18)	
Phase of underlying disease (%)			
Diagnosis	36 (97)	62 (95)	0.1
Relapse	1 (3)	3 (5)	
Type of chemotherapy (%)			
Ida/arac (3/7)	26 (70)	51 (78.5)	
Fludarabine-based			
Low dose Ara-C	11 (30)	12 (18.5)	0.3
Allogeneic stem cell transplant	0 (0)	2 (3)	
Quinolone prophylaxis (%)	12/37 (32)	20/65 (31)	0.9
Induction-1	12/37 (32)	50/65 (77)	0.001
Induction-2	2/7 (29)	9/11 (82)	0.07
Consolidation-1	5/26 (19)	33/38 (87)	0.001
Consolidation-2	4/13 (31)	18/20 (90)	0.001
Consolidation	2/4 (50)	12/12 (100)	0.08
Days of neutropenia (range)			
Induction-1	21 (17–27)	22 (19–28)	0.4
Induction-2	20 (16–33)	18 (17–24)	0.75
Consolidation-1	15 (15–19)	22 (13–27)	0.27
Consolidation-2	13 (11–18)	16 (13–22)	0.2
Consolidation3	21 (17–27)	15 (11–16)	0.14

**Table 2 jcm-08-01985-t002:** Isolated multidrug resistant Gram negative bacteria in surveillance cultures. ESBL: Extended Spectrum Beta-Lactamase producing. ESBL-CPN: Extended Spectrum Beta-Lactamase and carbapenemase producing. XDR: Extremely Drug Resistant.

Pathogen and Resistance Pattern *N* (% of Total Gram Negative Isolated)	*N* (%)
*N* of patients colonized/*n* of isolates	32/35
*Klebisella pneumoniae*, *n* isolates, % of isolates	23
• ESBL	20 (85)
• ESBL-CPN	3 (15)
*Pseudomonas aeruginosa*, *n* isolates, % of isolates	5
• XDR	1/5 (20)
• Non XDR (carbapenem resistant)	4/5 (80)
*Escherichia coli* ESBL, *n* isolates, % of isolates	1 (100)
*Enterobacter cloacae complex*, n isolates, % of isolates	4
• ESBL	1 (20)
• ESBL-CPN	3 (75)
*Citrobacter freundii complex* CPN, n isolates, % of isolates	1 (100)
*Acinetobacter baumannii XDR*, n isolates, % of isolates	1 (100)

**Table 3 jcm-08-01985-t003:** Types of Infections and univariate comparison between cohorts.

	Cohort 1	Cohort 2	Overall	*p*
No fever, no infection, n patients, number of cycles (%)	9/87 (10)	27/146 (18)	36/233(15)	0.9
Fever of Unknown Origin, n febrile episodes, (%)	21/89 (24)	50/147 (34)	71/236 (30)	0.9
Clinically documented, n febrile episodes, (%)	12/89 (12)	11/147 (7)	23/236 (10)	0.13
Catheter	0 (0)	1 (9)	1 (4)	0.79
Respiratory	4 (33)	5 (45)	9 (39)	0.67
Soft tissues	7 (58)	3 (25)	10 (44)	0.03
Gastrointestinal	1 (8)	2 (18)	3 (13)	0.8
Microbiologically documented, n febrile episodes, (%)	56/89 (63)	87/147 (59)	143/236 (61)	0.6
Gram negative	33 (59)	37 (43)	70 (49)	0.05
Gram positive	18 (32)	43 (49)	61 (43)	0.12
Viral disease	1 (1)	1 (1)	2 (1)	0.7
Fungi	2 (4)	4 (5)	6 (4)	0.8
Clostridium difficile	2 (4)	2 (2)	4 (3)	0.9

**Table 4 jcm-08-01985-t004:** Univariate comparisons of multi-resistant Gram negative bacteria incidence between cohorts, together with bacterial susceptibility pattern. Percentage is referred to bacteria of the same species recovered from cultures during the period. MR-GNB: multi-resistant Gram negative bacteria. ESBL: Extended Spectrum β-Lactamase producing. CPN: carbapenemase-producing. ESBL-CPN: Extended Spectrum β-Lactamase and carbapenemase-producing. XDR: Extremely Drug Resistant.

MR-GNB and Resistance Pattern N/Total GNB Recovered (% Resistant)	Cohort 1	Cohort 2	*p*
*Klebisella pneumoniae*			
• ESBL	1/8 (12)	11/18 (61)	0.06
• CPN	0/8 (0)	1/18 (6)	
• ESBL-CPN	7/8 (88)	6/18 (33)	
*Pseudomonas aeruginosa*			
• XDR	7/11 (36)	6/9 (55)	0.1
• Other	4/11 (36)	3/9 (34)	
*Escherichia coli*			
• ESBL	0/9	6/13 (46)	0.057
• CPN	1/9 (11)	0/13 (0)	
• Other	8/9 (89)	7/13 (54)	
*Enterobacter* sp. ESBL	0/3 (0)	1/4 (25)	0.8
*Stenotrophomonas maltophila*	1/1(100)	1/1 (100)	1

**Table 5 jcm-08-01985-t005:** Multivariate analysis of factors related to survival. Prophylactic fluoroquinolones and active surveillance were introduced in the model as an interaction.

Factor	Hazard Ratio	95% Confidence Interval	*p*
Age	1.005	0.98–1.03	0.68
Comorbidity index	2.025	0.88–4.63	0.09
Genetic risk	1.593	0.79–3.17	0.18
Prophylactic fluoroquinolones with no active surveillance	0.611	0.33–1.13	0.12
Prophylactic fluoroquinolones with active surveillance	0.555	0.38–0.79	0.001

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
