# Peer review of "The Value of Adding Surveillance Cultures to Fluoroquinolone Prophylaxis in the Management of Multiresistant Gram Negative Bacterial Infections in Acute Myeloid Leukemia"

_jcm, 2019, doi:10.3390/jcm8111985_

Round 1

Reviewer 1 Report

This study reports a clinically useful information/strategy for to improve overall survival of patients with AML. The authors found that fluroquinolone prophylaxis plus weekly surveillance cultures statistically significantly was associated with improved survival. I think this article, even being a single center with sample size, will be cited and used by many leukemia centers around the globe.

There are a few typos that need correction. The quality/arrangement of Tables 2 and 3 are quite confusing and need revision.

Supplementary Materials, Funding and Acknowledgments are highlighted in yellow with unanswered instruction for authors. These must be address.

Author Response

This study reports a clinically useful information/strategy for to improve overall survival of patients with AML. The authors found that fluroquinolone prophylaxis plus weekly surveillance cultures statistically significantly was associated with improved survival. I think this article, even being a single center with sample size, will be cited and used by many leukemia centers around the globe.

Thank you for your positive comments.

There are a few typos that need correction. The quality/arrangement of Tables 2 and 3 are quite confusing and need revision.

The manuscript has been reviewed and typos corrected. Tables 2 and 3 have been improved.

Supplementary Materials, Funding and Acknowledgments are highlighted in yellow with unanswered instruction for authors. These must be address.

The requested information has been added to the manuscript.

Reviewer 2 Report

Christelle and colleagues report on a prospective trial of adult patients with AML broken into two cohorts.  First cohort was a baseline group; the second cohort mandated fluoroquinolone antibiotic prophylaxis, routine screened for resistant bacteria by rectal swab, contact precautions instituted for those affected, and targeted therapy.

In summary, they evaluated 233 chemotherapy cycles in a total of 102 patients.  Patients were evaluated over multiple chemo cycles, and bmt.  Overall, the cohorts were similar, expect cohort 2 had more fluoroquinolone use, as mandated by protocol.

Findings included: a 50% colonization rate of drug resistant bacteria; those in cohort 2 had significantly less infectious mortality; and fluoroquinolone prophylaxis resulted in improved survival.

Overall, I think the paper is well written, and easy to follow. 

Strengths include well presented data, and very complete with appropriate highlighting of significant findings.

Limitations include relatively small numbers, and multiple subgroup analyses.

However, this paper provides excellent baseline data for patients with AML.  Active surveillance is not standardly done as outlined in this paper- and maybe it should be.

If desired, the discussion could be made more concise.  Paragraphs 4 and 6 of the discussion could be omitted or shortened.   

Author Response

Overall, I think the paper is well written, and easy to follow.

Thank you very much.

If desired, the discussion could be made more concise.  Paragraphs 4 and 6 of the discussion could be omitted or shortened.  

Paragraph 4 (now paragraph 5) has been shortened. Paragraph 6 has been removed, as some of its key concepts have been moved to prior paragraphs (now paragraph 3, line 272).

Reviewer 3 Report

There are some methodological and statistical issues in the manuscript. There is mention of which fluoroqinolone is used, when, day, duration, what dose etc. There is no univariate analysis described, only multivariate analysis. It should be completed and changes accordingly, if they do or not change the results. The groups were not really well matched and comparable, because the cohort 2 received 2-3 times more fluoroqinolone prophylaxis. Was the duration, kind of fluoroquinolone the same in both cohorts? Nothing of that I found in the methods.

I do not understand what was the reasons of increasing infections in consolidation therapy? The reasons why the G+ infections more in consolidation?

How the authors define contact precautions?

The conclusions - message to the readers should be more clear and straightforward, not reduced to statistical reasoning. What are authors saying that is new ?

What hade surveillance cultures added to the prophylaxis? 

Author Response

There are some methodological and statistical issues in the manuscript.

Thank you for your comments. We have addressed all your concerns regarding data analysis.

There is mention of which fluoroquinolone is used, when, day, duration, what dose etc.

Fluoroquinolone prophylaxis consisted on ciprofloxacin 500 mg PO q12 hours starting on day one from the chemotherapy cycle and continuing until absolute neutrophil count > 0.5 x 109/L. This has been clarified in the manuscript in page 2, line 80.

There is no univariate analysis described, only multivariate analysis. It should be completed and changes accordingly, if they do or not change the results.

We have added the following univariate comparisons between both cohorts:

page 3, line 130: rate of complete remission. page 4, table 1: rate of allogeneic stem cell transplant, demographic and leukemic variables, rate of fluroquinolone prophylaxis. Page 4, line 140: length of hospitalization. Page 6, table 3: Types of infection. Page 7, table 4: Univariate comparisons of multi-resistant Gram-negative bacteria incidence.

Additionally, we would like to clarify that univariate comparisons of rate of colonization were not possible, since surveillance cultures were not routinely performed in cohort 1.

The groups were not really well matched and comparable, because the cohort 2 received 2-3 times more fluoroquinolone prophylaxis. Was the duration, kind of fluoroquinolone the same in both cohorts? Nothing of that I found in the methods.

As the reviewer highlights, both groups were different regarding anti-infective prophylaxis. As indicated in Methods section, line 70, fluoroquinolone prophylaxis, although suggested, was left at the discretion of the responsible physician in patients allocated to cohort 1, resulting in low rates of prophylaxis (shown in Table 1). When indicated, FP was used in the same way as in cohort 2 and consisted on ciprofloxacin 500 mg PO q12 hours starting on day one from the chemotherapy cycle and continuing until absolute neutrophil count > 0.5 x 109/L.

On the contrary, the anti-infectious strategy in the second cohort, denominated active surveillance program, consisted on the following actions: (1) Mandatory fluoroquinolone prophylaxis from the beginning of chemotherapy; (2) rectal swabs for MR-GNB surveillance collection from the first day of hospital admission and weekly thereafter; (3) Contact precautions for MR-GNB colonized and infected patients; 4) targeted therapy for febrile neutropenia in colonized patients according microbiologic results of surveillance cultures. This has been also clarified in the manuscript in page 2, line 74 and forward. Therefore, rates of prophylaxis in this cohort increased up to 80% and above. This difference was expected, as it was part of the tested intervention, and has been clarified in line 141.

I do not understand what was the reasons of increasing infections in consolidation therapy?

Thank you for this interesting comment. To answer this question, we have calculated the rate of microbiologically documented infections occurring during induction and consolidation cycles by dividing the total number of infections per total number of chemotherapy cycles administered. Rate of infection was 1.05 and 2 for induction and consolidation cycles in cohort 1. In cohort 2, rate of infection was 0.89 during induction and 1.14 during consolidation. An increase rate of infections in consolidation has been observed previously, associated to the use of indwelling catheters and high dose arac, especially in patients without fluoroquinolone prophylaxis (Halim et al Ann Oncol 2017). To support this hypothesis, we must note that in our center, both the use of indwelling catheters and high dose arac in consolidation are standard approaches for the treatment of acute leukemia patients. Moreover, the higher rate of infection/cycle during consolidation in cohort 1 compared to cohort 2 may be a consequence to the infrequent use of fluoroquinolone prophylaxis in this cohort (table 1). These issues have been clarified in the Results section (page 6 paragraphs 1 and 2) and in the Discussion section (page 10, line 274 and forward).

The reasons why the G+ infections more in consolidation?

This is an interesting point. We speculate that the systematic use of fluroquinolones is correlated with the increasing incidence of Gram positive infections during consolidation and a shift from Gram negative towards Gram positive infections.

These results are in accordance with previous reports: in a study performed on 200 acute leukemia patients treated with ciprofloxacin prophylaxis, Saini et al reported that gram-positive organisms constituted 79% and 70% of the isolates during consolidation cycles 1 and 2, respectively (Saini BMC infect Dis 2013). Likewise, Halim et al (Ann Oncol 2017) reported a “shift from Gram-negative to Gram-positive likely due to the introduction of ciprofloxacin prophylaxis, in particular in ambulatory care patients treated with ciprofloxacin prophylaxis compared with inpatients without ciprofloxacin prophylaxis”. Another reason that explains this shift is the use of high dose arac in consolidation in comparison with the standard regimen 3/7 used in induction, which causes oral and gastrointestinal damage resulting in infection. This is also the practice in our center: a double induction regimen with 3/7 followed by sequential courses of high dose Ara-C.

We have introduced two lines highlighting this finding in the Results section (page 6, line 180) and discussed it in the Discussion section, page 9, line 248 and Page 10, line 274.

How the authors define contact precautions?

Contact precautions for MR-GNB infected patients included patient placement (single-room), gowns, gloves, noncritical patient-care equipment/patient-dedicated use of such equipment, enhanced environmental measures (water filters and cleaning) and antiseptic baths. This has been described in the methods section (line 74).

The conclusions - message to the readers should be more clear and straightforward, not reduced to statistical reasoning. What are authors saying that is new ?

We have improved the conclusions of our manuscript to include a clear message. Therefore, the following sentence has been added in the conclusions section: “Our results show that for FP to be effective in AML patients, it must be included in a set of measures including systematic surveillance cultures and isolation protocols. Microbiological data obtained from surveillance cultures allows the early onset of effective contact precautions and targeting therapy in patients with multi-resistant Gram negative bacteria. This strategy must be continued throughout every chemotherapy cycle and during stem cell transplantation.”

What have surveillance cultures added to the prophylaxis?

As discussed, surveillance cultures allow the early isolation of colonized patients to limit the spread of MR bacteria and the onset of effective, targeted antimicrobial agents. Moreover, this can help to avoid the overuse of broad spectrum antibiotics in non-colonized patients. These benefits are discussed in paragraphs 5 and 6 of the Discussion.

Round 2

Reviewer 3 Report

The authors answered with relevant comments to the questions.